# African Cultivated, Wild and Weedy Rice (*Oryza* spp.): Anticipating Further Genomic Studies

**DOI:** 10.3390/biology13090697

**Published:** 2024-09-05

**Authors:** Babatunde O. Kehinde, Lingjuan Xie, Beng-Kah Song, Xiaoming Zheng, Longjiang Fan

**Affiliations:** 1Institute of Crop Science, Institute of Bioinformatics, Zhejiang University, Hangzhou 310058, China; kbabatundeo2@gmail.com (B.O.K.); xielingjuan@zju.edu.cn (L.X.); 2Department of Zoology, University of Lagos, Akoka-Yaba, Lagos 101245, Nigeria; 3School of Science, Monash University Malaysia, Bandar Sunway 46150, Selangor, Malaysia; song.beng.kah@monash.edu; 4Yazhouwan National Laboratory, Yazhou District, Sanya 572024, China; zhengxiaoming@caas.cn

**Keywords:** African rice, *Oryza glaberrima*, NERICA, weedy rice, genome sequencing

## Abstract

**Simple Summary:**

1. Three types of rice grown in Africa: The native African rice *O. glaberrima*, the Asian rice (*O. sativa*) and the new type of African rice (e.g., NERICA varieties). *O. sativa* has mostly supplanted *O. glaberrima* as the dominant crop in Africa. 2. There is no evidence of the existence of Asian weedy rice (*O. sativa* f. *spontanea*) in Africa, while African weedy rice (*Oryza* spp.) is a complex population which harbors both a native and Asian genetic background. The genomic approach is an ideal way to characterize the African weedy rice population structure. 3. There have been no genomics studies on African weedy rice yet, and the latest genomic study on African cultivated rice occurred five years ago.

**Abstract:**

Rice is a staple crop in sub-Saharan Africa, and it is mostly produced by Asian cultivars of *Oryza sativa* that were introduced to the continent around the fifteenth or sixteenth century. *O. glaberrima*, the native African rice, has also been planted due to its valuable traits of insect and drought tolerance. Due to competition and resistance evolution, weedy rice has evolved from *O. sativa* and *O. glaberrima*, posing an increasing threat to rice production. This paper provides an overview of current knowledge on the introduction and domestication history of cultivated rice in Africa, as well as the genetic properties of African weedy rice that invades paddy fields. Recent developments in genome sequencing have made it possible to uncover findings about *O. glaberrima*’s population structure, stress resilience genes, and domestication bottleneck. Future rice genomic research in Africa should prioritize producing more high-quality reference genomes, quantifying the impact of crop–wild hybridization, elucidating weed adaptation mechanisms through resequencing, and establishing a connection between genomic variation and stress tolerance phenotypes to accelerate breeding efforts.

## 1. Introduction

Rice has emerged as a crucial staple food crop in sub-Saharan Africa (SSA) over the last 20 years, becoming integral to both food security and income generation across the continent [1]. The most widely cultivated rice species in the world belong to two cultivated types, namely Asian cultivated rice (*Oryza sativa* L.) and African cultivated rice (*Oryza glaberrima* Steud.). The former, which is one of the world’s major crops, feeds half of the world’s population. Overall, rice has become a commodity of strategic importance in many African countries [2], although rice farming on the continent has a relatively short history compared to Asia, where rice cultivation originated thousands of years ago. Among the food sources on the continent, rice is growing at the highest rate due to significant urbanization in Africa, surpassing any other region in the world [3].

Weedy rice is one of the most problematic weeds and a threat to global rice production [4]. The emergence of weedy rice occurred through processes including de-domestication, hybridization with sympatrically growing wild relatives of *Oryza*, and selection within rice cropping systems [5]. Therefore, alongside native wild rice such as *O. barthii* and *O. longistaminata*, as well as the diverse cultivated rice, the African continent presents a unique ecological environment and complex genetic background for rice. This environment could make it easier for weedy rice to emerge in Africa, posing significant challenges to rice production due to its competitive nature and similarity to cultivated rice.

Since the first sequencing of the rice genome in 2002 [6,7], significant genomic studies have been conducted on Asian cultivated rice [8,9,10,11,12,13,14]. In addition, many populations of Asian weedy rice were recently investigated through genome resequencing [15,16,17,18]. However, the latest genomic study on African cultivated rice occurred five years ago. African weedy rice, which is also an important genetic resource, has been neglected, with no studies on its genomics having been conducted. Most of the genomic characteristics of African weedy rice have still not been characterized in detail, and the genetic basis of their evolutionary history has rarely been studied. Genomic studies offer immense potential to address these challenges. By unraveling the genetic basis of important traits in African rice species and their weedy relatives, we can develop more resilient, higher-yielding varieties and implement more effective weed management strategies. Reviewing the evolution, challenges, and adaptations of African diverse weedy rice highlights important research gaps for creating high-yielding rice varieties and integrated weed management strategies.

In this review, we focus on studies on African cultivated, wild, and weedy rice (*Oryza* spp.) from a genomic perspective by providing a synthesis of the current understanding regarding the following two key issues: (1) the introduction and evolution of the cultivated Asian rice in Africa and (2) the effects this may have had on the genetic makeup and competitiveness of African weedy rice populations. Additionally, we identify key research gaps and highlight future directions that could significantly impact African rice breeding and cultivation practices.

## 2. Overview of African Cultivated and Wild Rice

### 2.1. The Introduction of Asian Rice (O. sativa)

Asian rice was introduced to West Africa by the Portuguese in about 1500 AD, and some historians have speculated that Arab traders may have transported *O. sativa* from Southeast Asia or India into East Africa via marine routes across the Indian Ocean [19,20]. It is believed that rice from Asia migrated to Africa by a number of different routes at different times [21], and expanding trade networks between Asia and areas of East and West Africa are thought to have contributed to the introduction of *O. sativa* to Africa via two potential marine routes in the fifteenth and sixteenth centuries [22]. Beye et al. [23] in their study mentioned that Asian rice arrived in Africa through Madagascar due to human migration and Indonesian colonization. According to Bezançon [24], the European maritime explorers and traders are credited with being the ones who brought Asian rice to the coasts of West Africa prior to the modern era, and that they would have done so while returning from Asia between the end of the fifteenth and the beginning of the sixteenth century (Figure 1). However, the exact details, timing, and routes of *O. sativa*’s initial introduction to Africa remain uncertain. What is known is that its introduction occurred sometime around the fifteenth or sixteenth century and that active maritime trade networks during this period provided the means for Asian rice varieties to make their way from Asian hubs of rice cultivation to coastal African port cities [22].

### 2.2. African Cultivated and Wild Rice

Based on FAOSTAT, rice is mainly planted in Sub-Saharan Africa (SSA). As shown in the stated countries (Nigeria, Tanzania, Mali, Ghana, Burkina Faso, Uganda, Ethiopia, Kenya, Niger, and Zambia) between 2016 and 2020, the influence of this transition is evident in important measures, including the average national rice output, harvested area, average yield, and the percentage of rice cultivation under rainfed conditions (Table 1). Prior to the introduction of *O. sativa* from Asia in the fifteenth and sixteenth centuries, the only cultivated species found in Africa was *O. glaberrima*, which is valued for its resilience in challenging conditions, including drought tolerance and pest resistance. Currently, these two types of rice are cultivated in Africa, with *O. glaberrima* being grown on a smaller scale. Though efforts have been made to preserve *O. glaberrima*, *O. sativa* has largely replaced *O. glaberrima* as the primary cultivated rice in Africa since its introduction due to higher yields and other agronomically desirable traits [22]. Meanwhile, some new type of varieties, produced by the interspecific hybridization of *O. sativa* and *O. glaberrima*, such as the New Rice for Africa (NERICA) varieties, have been introduced and widely adopted by farmers in many parts of Africa (https://www.africarice.org/nerica accessed on 17 March 2024).

Asian rice is now the most common variety in Africa. However, its proportion in the continent’s rice production varies widely across regions, from less than 50% to nearly 100%. Ghesquière [30] posited that *O. glaberrima*, which was once widely cultivated, is being supplanted in African fields by *O. sativa* due to its low yield, lodging and shattering vulnerability, and the introduction of high-yielding cultivars. However, smallholder rice farmers in some West African countries continue to cultivate the crop despite its decline [31,32]. The African rice farming sector experienced significant changes with the introduction of Asian rice (*O. sativa*). This shift is noticeable in specific regions where the cultivation of the native African rice, *O. glaberrima*, has been supplanted.

In addition to the two cultivated rice species, Africa is also home to important wild genetic resources, such as *O. barthii* and *O. longistaminata*, which have been exploited to enhance cultivated varieties through breeding efforts [33]. Other wild rice species containing valuable traits are BB (*O. punctata*), BBCC (*O. schweinfurthiana*), CC (*O. eichingeri*), and FF (*O. brachyantha*) [34]. Since these species span 10–15 million years of evolutionary history, Africa plays an important role in understanding and preserving the genetic diversity of the *Oryza* genus [35]. Wild species like *O. nivara*, *O. rufipogon*, and *O. longistaminata* possess the same genome ‘AA’ as cultivated rice, allowing for easy crossbreeding with the cultivated *O. sativa* and *O. glaberrima* species. Studies have established that *O. glaberrima* (African rice), is native to West Africa and that it was domesticated from its wild progenitor species, *O. barthii*, formerly known as *O. breviligulata* [31].

**Table 1 biology-13-00697-t001:** National average total rice production, harvested area, and average yield for selected African countries in 2016–2020. Also shown is the percentage of area under rainfed conditions, including both lowland and upland environments (FAOSTAT, 2022; [36]).

Country	Total Production (Thousand ton)	Harvested Area (Thousand ha)	Average Yield (t/ha)	Rainfed Rice (as % of Total Rice Area)
Nigeria	8080	5409	1.50	99
Tanzania	3220	1162	2.77	93
Mali	2972	879	3.39	37
Ghana	815	285	2.85	82
Burkina Faso	367	171	2.14	92
Uganda	212	77	2.76	42
Ethiopia	164	57	2.87	99
Kenya	127	28	4.66	<1
Niger	124	28	4.37	100
Zambia	34	26	1.31	62

### 2.3. Interspecific Hybridization: The New Rice for Africa (NERICA) Varieties

While most rice varieties grown in Africa were originally developed in Asia, great breeding efforts have led to the development and release of regionally adapted varieties of *O. sativa* since the 1960s. One such variety is the NERICA (New Rice for Africa) variety, which combines the high yields of Asian rice (*O. sativa*) with the stress tolerance and adaptation of African rice (*O. glaberrima*) [37,38]. By crossing introgression restorer lines carrying *O. glaberrima* genes with the sterility lines of *O. sativa*, a partially interspecific hybrid rice has been developed. A positive interspecific heterosis for grain yield has been demonstrated by these hybrids [39]. The first generation of 18 NERICA lines are adapted to the upland rice ecology across sub-Saharan Africa (SSA), and some have yield advantages over parent varieties through drought tolerance, pest/disease resistance, and higher yield. Additionally, NERICA rice has been tested and resulted in higher grain protein content across Africa, most significantly in areas under upland production, which improves nutrition and food security [40].

There are now a total of 82 NERICA varieties, including 18 varieties for the upland ecology, 60 rainfed lowland varieties, and 4 irrigated varieties (www.africarice.org/nerica accessed on 17 March 2024). Other varieties that are suited to African agroecologies and production restrictions are FARO 66, FARO 67, FARO 52, WAB 56-50, Sahel 108 (from IR8), WAB 638-1 (from IR64), ITA306, etc. [41], and creating better locally adapted cultivars is still a key priority if rice productivity is to continue rising throughout Africa. Furthermore, breeders in Bangladesh, China, India, and several other countries are utilizing NERICA cultivars in varietal improvement programs (https://www.africarice.org/nerica accessed on 17 March 2024). For the Aus season, the Bangladesh Institute of Nuclear Agriculture and the Bangladesh Rice Research Institute, for instance, developed and released three drought-tolerant rice varieties from NERICA-10, namely BRRI dhan 82, BINA dhan 19, and BINA dhan 21 [42].

## 3. African Weedy Rice

### 3.1. Weedy Rice: Origins and Traits

Weedy rice (*O. sativa* spp.) is a widespread weed that infests areas in and around rice fields [17,43]. It is known as *O. sativa* L. and *O. sativa* f. *spontanea* when referring to rice from Asia, but it is only referred to with these terms in the African context when the lineage of the referenced weedy population is known [44]. Regarding the origin of weedy rice, several hypotheses have been proposed [5]. These hypotheses encompass the idea that weedy rice could be an evolved taxon that descended from cultivated rice or that it could be a crop mimic that resembles a wild relative of rice. They could also be the result of natural crossings between wild and cultivated rice [45]. The origins of weedy rice in Asia and West Africa differ from those in the Americas or Europe due to their status of being the cradles of evolution and domestication for the two cultivated rice species (*O. sativa* in Asia and *O. glaberrima* in Africa). These regions have been cultivating rice for thousands of years, and they are home to a variety of wild rice species, many of which have evolved into problematic weeds. To effectively manage the global impact of weedy rice on rice agriculture, it is imperative to comprehend its origins and characteristics [46].

### 3.2. Phylogenetic Relationships: Connecting Weedy, Cultivated, and Wild Rice

The cultivated rice varieties, *O. sativa* and *O. glaberrima*, are closely related to various weedy rice forms. These weedy plants share several characteristics with the two main cultivated species. Wild species like *O. nivara*, *O. rufipogon*, and *O. longistaminata* have the same ‘AA’ genome as cultivated rice, which allows for easy crossbreeding with *O. sativa* and *O. glaberrima* (Figure 2). The genomic similarity between cultivated rice varieties and weedy forms enables hybridization and genetic exchange [46]. A comprehensive genomic analysis has identified five subpopulations of African wild rice, among which *O. glaberrima* exclusively clusters in specific regions [31]. Therefore, unlike Asian weedy rice, which originates from cultivars of *O. rufipogon* or *O. sativa*, African weedy rice is determined to be a genetically distinct hybrid of local wild and cultivated species [47]. According to Nuijten [48], one unintentional factor contributing to this challenge is the existence of weedy rice types exhibiting characteristics that are intermediate between wild African rice (*O. barthii*) and *O. glaberrima* in farmers’ fields, a situation that raises concerns about gene flow between weedy types and cultivated Asian rice, which could result in in-field interspecific hybridization. It is notable that *O. glaberrima* is believed to have originated from the wild species *O. barthii*, also known as *O. breviligulata* [46]. This indicates the complex relationships that exist within the *Oryza* genus as well as the possibility of gene flow and hybridization among these related species. It also suggests a close evolutionary relationship between weedy rice and the wild species from which cultivated varieties originated. To effectively control and mitigate the adverse impacts of weedy rice on cultivated rice varieties, it is essential to comprehend these phylogenetic relationships between cultivated rice, weedy rice, and wild rice species.

### 3.3. Genetics and Adaptation of African Weedy Rice

Weedy rice possesses several adaptive traits which contribute to its diversification and productivity, such as seed shattering, seed dormancy, prolonged emergence, and pigmented seed pericarp in many cases [50]. The weedy rice accession B2 showed high allelopathy and herbicide tolerance, characteristics that could be valuable for rice improvement efforts [51]. Stress tolerance genes help weedy rice withstand various environmental challenges [52], while unknown regulatory genes, different from those in wild rice, contribute to the earlier seed shattering observed in weedy rice, thereby enhancing its fitness for survival in the environment [53,54]. Based on the diversification and productivity posed by these traits, African weedy rice may have a high concentration of genes associated with stress tolerance, allelopathy, and seed shattering. These traits may also contribute to African weedy rice’s resilience and abundance by improving its ability to survive and reproduce in agricultural environments. Previous research has suggested that about 55% of the rice fields in Senegal, West Africa, are reported to be affected with weedy rice biotypes [55,56].

### 3.4. Genomic Studies on African Native Rice

Genomic research on African rice has provided insights into its genetic diversity and domestication history [57,58,59]. Initial studies by Cubry [29] suggest the independent domestication of African rice (*O. glaberrima*) in the Inland Delta of the Upper Niger River. Further investigations by Wang and Cubry [30,60] have identified genes associated with domestication traits, such as seed shattering, seed size, plant architecture, flowering time, and resistance to Rice Yellow Mottle Virus. However, recent studies by Choi and Veltman [25,61] challenge the conventional understanding of African rice domestication, proposing a non-centric or polycentric view. This suggests a more complex domestication process involving several regions [62]. Comparative genomic studies between African and Asian rice by Huang and Ma [63,64] have identified regions of strong divergence and shared selection targets during domestication. Organelle-to-nucleus DNA transfers have also been discovered to influence nuclear genome divergence [64,65]. Understanding the population structure of African rice has been crucial in elucidating its domestication history. Studies by Veltman, Huang, Li, Nabholz, Ndjiondjop, Orjuela, Semon, and Wambugu [47,61,63,66,67,68,69,70] suggest a largely homogeneous population with minimal partitioning. African wild rice species, such as *Oryza longistaminata*, *Oryza barthii*, *Oryza punctata*, *Oryza brachyantha*, and *Oryza eichingeri*, exhibit extensive genomic diversity [58,62,71,72,73,74,75,76,77,78,79,80]. Among these species, *O. longistaminata* stands out for its beneficial traits, and genomic analysis has unveiled crucial molecular mechanisms [72,73,79,81]. Additionally, studies have established a close morphological and genetic relationship between *O. barthii* and African rice, suggesting shared ancestry and a genomic similarity [31,57,58,61,63,76,82]. These genomic insights contribute significantly to understanding the genetic diversity, domestication histories, and adaptive mechanisms of native African rice and wild rice species, providing pathways for the development of improved rice varieties (see Table 2).

## 4. Challenges and Future Directions

### 4.1. Mystery of African Weedy Rice

Uncertainty surrounds the origins of weedy rice in Africa. Possible explanations include the de-domestication of native African rice, hybridization between African and Asian rice, the de-domestication of introduced Asian rice, and the de-domestication of NERICA (an African–Asian hybrid rice) (Figure 3). Asian rice genes were probably integrated into African weedy rice by gene flow between wild African rice and cultivated Asian rice. The complex genetics of African weedy rice may be explained by the numerous historical introductions of Asian rice to Africa. Weedy rice varieties like “ngewobei” and “ngafabei” may act as a link between wild and cultivated rice species, enabling breeders to transfer beneficial characteristics from wild to cultivated varieties. However, solving the problems posed by weedy rice requires considering its diverse forms and the potential for hybridization with cultivated rice [48]. Furthermore, the history of human migration, trade, and exchanges in Africa may have influenced the diversity of weedy rice through the periodic introduction of new Asian rice varieties. Unintentional selection probably promoted weedy characteristics through agricultural practices and human selection [86]. Due to variations in functional traits, weedy rice populations show greater adaptability to a wide range of environmental conditions, making them challenging to control [87]. It is evident that the evolution of weedy rice in Africa is a complex and diverse phenomenon when one considers the various aspects of interactions between Asian and African rice, human activities, environmental factors, and genetic processes. To solve the mystery surrounding the origin and dynamics of weedy rice populations in Africa, a comprehensive approach taking all these factors into account is essential.

### 4.2. Africa Needs More Genomic Studies on Local Rice

The global plant community believes that plant genomics research in Africa holds great potential for crop improvement and the utilization of medicinal plants [88]. African plant genome sequencing will provide a fundamental framework for the development of regional omics techniques, with emphasis on microbiomes, metabolites, and genomes. African scientists must acquire the knowledge and resources needed to promote high-quality plant genomics research aimed at resolving the continent’s agricultural challenges [88]. For example, there are no genomic studies yet on (1) pan-genomes and high-quality genomes, such as the T2T genome of African native rice, which are the basis of genetic resource mining; (2) African weedy rice, for which population genomics will provide key evidence for their origin and phylogeny.

### 4.3. Challenges of Genomic Studies in Africa

A key barrier to the implementation of agricultural genomic research in Africa is the lack of resources, both in regard to human and institutional capacities as well as funding [89]. Based on projections, genetic data may become the world’s largest data generator by 2025 [90], with existing repositories like NCBI GenBank, ENA, and DDBJ [91] facilitating global collaboration. However, the absence of an African replicated database poses challenges, and emphasizing the need for an ‘African Nucleotide Repository’ synchronized with international databases is crucial, necessitating increased training for researchers to improve data accessibility and utilization [89].

The utilization of preserved genetic resources faces challenges due to the lack of evaluation data [92,93], which is specifically evident in under-characterized and underutilized African *Oryza* in rice breeding programs. Concerted efforts in regard to sharing genebank accession-level information are crucial, especially with the decreasing cost of next-generation sequencing enabling large-scale genomic analyses [94]. Expanding high-quality assemblies is imperative for in-depth structural and functional genomic analysis, particularly concerning the largely uncharacterized African weedy rice populations compared to cultivated varieties. Understanding the evolutionary history, weed adaptation mechanisms, and genetic basis of traits is vital for exploiting genetic diversity [95].

There is some contention about whether the domestication process of *O. glaberrima* was truly independent of Asian rice [96], but the former experienced a drastic domestication bottleneck. This indicates the opportunity for African breeders to exploit the genetic diversity of its closely related wild ancestor [95]. Urgent resequencing studies on weedy populations in various African agroecologies are needed to extensively study the proportional introgression of crop–weed–wild breeds and control the weediness threat. High-density genotyping using genotype-by-sequencing techniques can detect introgressions from other rice species or wild ancestors that are significant to the region [95,96].

Construction of the connections between genetic variations and phenotypic traits is crucial for applying genomics in plant improvement. These relationships are a key focus in both applied plant improvement and fundamental biological research. The development of representative genome-wide variant datasets capturing African diversity will enable more potent genome-wide association studies, linking SNPs to stress resilience and regional adaptation. Addressing these constraints holds the possibility for a significant impact on African rice breeding and research [95].

## 5. Ethical and Social Implications of Rice Genomics in Africa

### 5.1. Current State of Rice Genomics in Africa

Conventional breeding techniques have led to considerable progress in developing modern rice varieties with higher yield potential, better grain quality, and improved resilience [97]. However, these techniques typically take 10–15 years due to the complexity of important agronomic traits and the need to eliminate undesirable genes from donor parents [98]. Genome editing can significantly accelerate this process, potentially reducing the time required by nearly two-thirds [99]. These methods enable scientists to introduce mutations with high precision, resolve problems encountered in conventional breeding, and allow for simultaneous editing of several genes [100].

An increasing number of African countries have shown policy interest in gene-editing technologies to address food security, reduce imports, and increase agricultural competitiveness [101,102]. Countries like South Africa, Sudan, Ethiopia, Ghana, Burkina Faso, and Zimbabwe are working towards developing national biosafety guidelines for regulating gene editing and GM crops as part of their food self-sufficiency plans [103]. However, there are sociopolitical challenges in fully embracing genome-edited innovations, with most African countries progressing slowly in implementing functional regulatory frameworks [104,105].

### 5.2. Ethical and Social Considerations

Recent research demonstrates that risks associated with gene editing are comparable to those of conventional breeding methods [99,103]. Nevertheless, critical scholars and advocacy groups raise concerns about potential physical off-target hazards from gene editing, as well as socioeconomic issues including loss of food and seed sovereignty, entrenchment of intensive agriculture and corporate power through patent regimes, and little consideration for local farming practices and conditions [106,107].

The socioeconomic impacts of adopting new rice varieties can be complex. While improved varieties can increase yields and farmer incomes, they may necessitate adjustments in farming practices or inputs. Smallholder farmers must not be marginalized by technologies that favor large-scale production. Furthermore, the potential loss of traditional varieties as farmers adopt improved lines could impact cultural practices and dietary diversity.

Intellectual property rights are a major concern in rice genomics. Many genomic technologies and elite germplasms are protected by patents owned by institutions in developed countries. Establishing equitable partnerships and ensuring that African researchers have the freedom to use these technologies is crucial. Open-source breeding initiatives and public-private partnerships could help address these challenges.

### 5.3. Responsible Research and Innovation Framework

The Responsible Research and Innovation (RRI) framework can guide the responsible design of technological innovations based on the principles of inclusiveness, anticipation, reflexivity, and responsiveness [108,109]. RRI stresses the early engagement of diverse stakeholders, the anticipation of potential social and ethical impacts, fostering reflexivity in regard to broader goals and assumptions, and being responsive to changing situations based on stakeholder feedback [110,111].

### 5.4. Strategies for Implementing RRI in African Rice Genomics

Implementing RRI in African rice genomics requires a multifaceted approach addressing scientific, social, ethical, and economic considerations. Key strategies include participatory variety selection, establishing farmer field schools, collaborating with local seed systems, and preserving traditional varieties and farming practices. Integrating traditional knowledge into breeding programs recognizes farmers’ experiential wisdom and can lead to more locally adapted varieties.

Robust regulatory frameworks are necessary for the approval, commercialization, and monitoring of CRISPR-based crops [103]. Technology transfer and capacity-building initiatives are essential for empowering local researchers [112]. Ethical considerations should guide CRISPR technology application, especially when editing genes with potential human health implications [113].

Public engagement and stakeholder involvement in decision-making processes are crucial. Countries must assess how CRISPR-edited crops align with international trade agreements and market access requirements [114]. Addressing potential impacts on biodiversity, ownership of genetic resources, and socioeconomic implications through participatory approaches is essential [115].

A case study in Madagascar on CRISPR-Cas9 for rainfed rice varieties highlights the need for transparent communication and reflection on effects on agricultural biodiversity and farming practices [116].

### 5.5. Challenges and Opportunities

Implementing RRI frameworks in African contexts may face challenges due to limited institutional capacities, budgetary constraints, and fewer researchers. International partnerships can address some of these gaps by collaborating with grassroots organizations and community groups [117].

While many rice studies using gene-editing tools are still in early laboratory stages, there is an opportunity to incorporate inclusive and reflexive practices early in technology development. This can involve working with smallholder farmers to identify key priorities, test gene-edited varieties during field trials, and engage institutional structures to build local capacities for gene editing [118].

## 6. Conclusions

The knowledge of African rice cultivation has expanded significantly in recent years due to advancements in genomic technologies and increased research focus. The intricate nature of rice cultivation is highlighted by the complex interactions that exist between native African rice, introduced Asian cultivars, and emerging weedy rice populations. The creation of adaptable and sustainable rice cultivation systems is urgently needed due to changing climates and evolving agricultural practices.

The genomic era offers unprecedented opportunities for African rice improvement, enabling the development of varieties combining high yield potential with stress tolerance and local adaptation. Practical applications of these genomic studies could substantially impact African agriculture through marker-assisted selection, accelerating the development of disease-resistant and stress-tolerant varieties adapted to diverse African agroecologies. Genomics-assisted breeding could also improve nutritional profiles and inform targeted weed management strategies.

A comprehensive breeding strategy should integrate genomic selection, marker-assisted selection, and genetic engineering or gene editing while implementing participatory breeding programs. We propose establishing a pan-African rice genomics network, developing diverse accession panels for high-resolution genome-wide association studies, and investing in capacity building for genomics and bioinformatics in African institutions. However, ethical and social considerations must be addressed. Implementing a Responsible Research and Innovation framework is crucial, emphasizing inclusiveness, anticipation of impacts, and responsiveness to stakeholder feedback. This approach should balance technological advancements with considerations of intellectual property rights and preservation of traditional varieties.

Future research priorities include producing high-quality reference genomes, quantifying crop–wild hybridization impacts, elucidating weed adaptation mechanisms, and connecting genomic variations to stress tolerance phenotypes. By combining advanced genomic technologies with traditional methods and local knowledge, we can develop rice varieties that are high-yielding, resilient, and appropriate for African contexts. This multifaceted approach, coupled with sustained investment and stakeholder engagement, will be crucial in realizing genomics’ full potential for African rice improvement. The potential impact on food security and economic development is substantial, promising a more food-secure future for the continent through increased yields, enhanced resilience to climate change, and reduced losses to pests and diseases.

## Figures and Tables

**Figure 1 biology-13-00697-f001:**
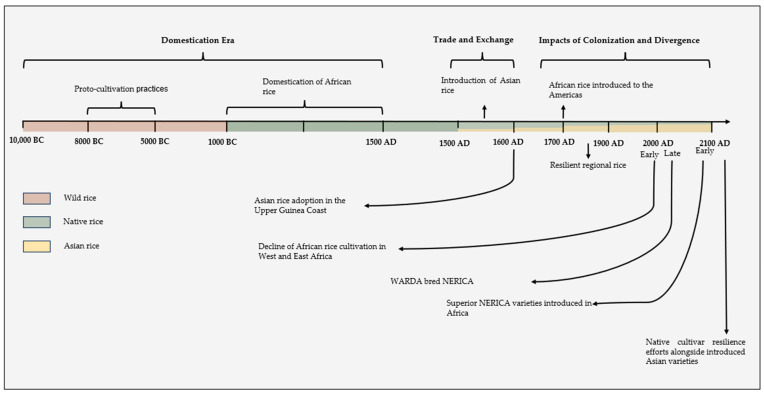
History of African rice, including the domestication of native African rice and the introduction of Asian rice, Introduction of Asian rice [22], Asian rice adoption in the Upper Guinea Coast [22], Resilient regional rice [22], Decline of African rice cultivation in West and East African, [22], Domestication of African rice [25], African rice introduced to the Americans [26], WARD bread NERICA [27], Superior NERICA varieties introduced in Africa [28], Native cultivar resilience efforts alongside introduced Asian varieties [29].

**Figure 2 biology-13-00697-f002:**
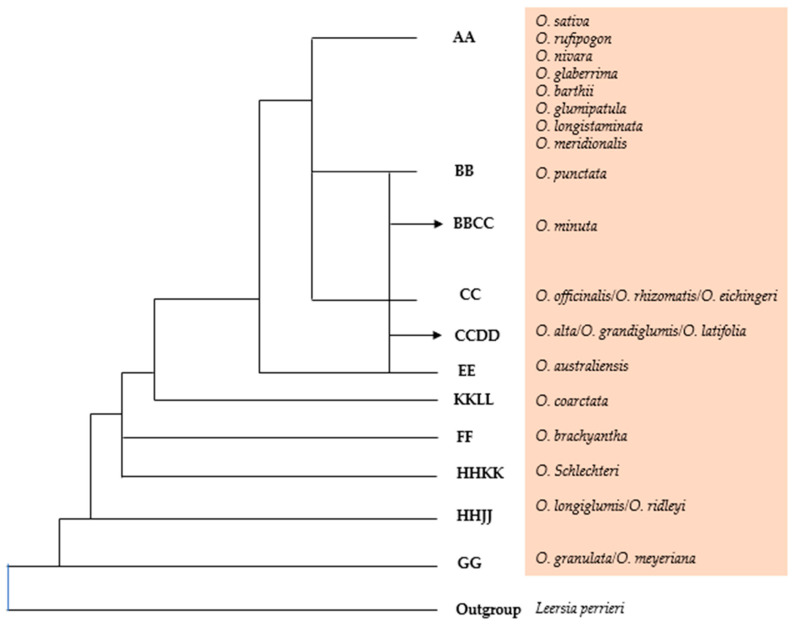
Phylogenetic relationship among the species of the genus *Oryza* [49].

**Figure 3 biology-13-00697-f003:**
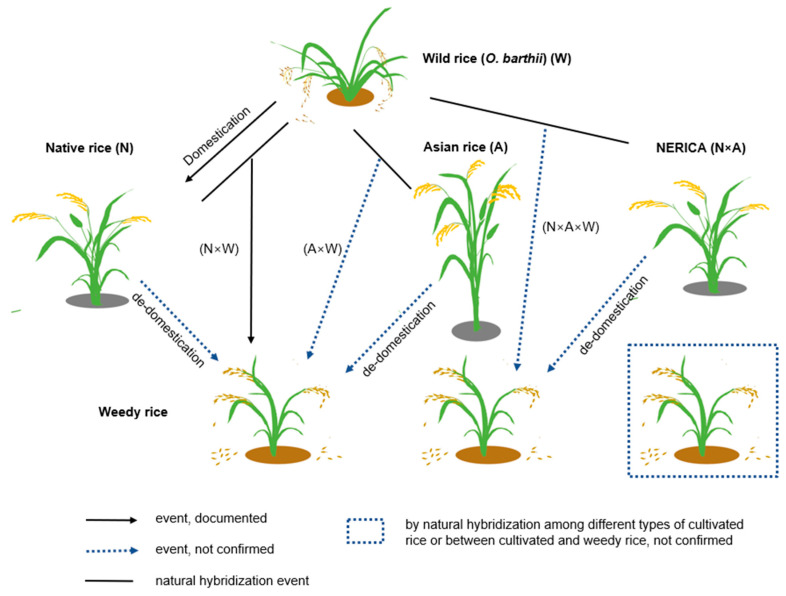
The potential origins of weedy rice in African paddy fields.

**Table 2 biology-13-00697-t002:** Summary of genomic studies on African native rice.

Species	Methodology	References
*O. longistaminata*	Using a hybrid Illumina and PacBio sequencing approach, a high-quality assembly of African wild rice genome was generated, resulting in a 363.5 Mb assembly with long scaffolds that were anchored into 12 pseudo-chromosomes.	[72]
*O. glaberrima*	A total of 163 resequenced *O. glaberrima* accessions were used for a genome-wide association study utilizing complementary statistical association methods to examine traits such as flowering time, panicle architecture, Rice Yellow Mottle Virus resistance, and climatic adaptation.	[60]
*O. barthii*, *O. glaberrima*, *O. rufipogon*, *O. nivara*	Four *Oryza* species were sequenced via PacBio and next-generation sequencing, assembled, and compared to identify organelle-to-nucleus DNA transfers, detect structural variations, and analyze the distribution of giant nuclear integrants in various rice populations.	[83]
*O. glaberrima*	Whole-genome resequencing involved analyzing 282 individuals with ~16.5× for both domesticated and wild African rice samples. Estimating genetic diversity, population structure, phylogenetic linkages, and admixture were all included in the analyses of population genetics.	[25]
*O. glaberrima*	Using phylogenetics, geographic modeling, and selective sweep detection, population genetics analysis enabled the whole-genome sequencing of 100 *O. glaberrima* accessions at ~10× coverage.	[61]
*O. longistaminata*	By utilizing synteny with *O. sativa* and a high-density linkage map, the genome of *O. longistaminata* was assembled into chromosomes after being sequenced using SMRT sequencing. A reference genome of high quality was validated using gene annotation and quality assessment.	[79]
*O. glaberrima*	Sequencing 246 genomes covering a wide geographic range across Africa, with ~37× coverage. Population genetics analyses were performed to assess genetic diversity, population structure, linkage disequilibrium, etc.	[29]
*O. glaberrima*, *O. longistaminata*, *O. punctata*, *O. glumaepatula*, *O. barthii*, etc.	The 13-genome dataset was generated using either long-read or short-read technologies, with robust scaffold support derived from long-insert library reads, including BAC-ends.	[57]
*O. glaberrima*	Whole-genome sequencing of three *O. glaberrima* accessions (CG14, Tog5680, and IRGC100991) at ~10× coverage, followed by de novo assembly.	[84]
*O. glaberrima*	Analyses of conserved 2179 *O. glaberrima* accessions using 27,560 DArTseq-based SNPs. Principal component analysis and model-based structural analysis to analyze population structure, a mini-core collection.	[68]
*O. glaberrima*	Genome sequencing and assembly of *O. glaberrima* accessions. Population structure, genetic diversity, selective sweeps, domestication genes, and evidence of the independent domestication of African rice were analyzed using the genome and SNP data.	[31]
*O. glaberrima*	The genome of *O. glaberrima* ~220 Mb was generated by shotgun sequencing utilizing subtractive hybridization and methylation filtration for gene enrichment. African and Asian rice were found to have different SSR marker polymorphisms, splice site substitutions, and species-specific genes when compared to *O. sativa*.	[85]
*O. glaberrima*, *O. barthii*	Sequencing of 14 unlinked nuclear genes in 20 cultivated and 20 wild rice accessions, with analyses of nucleotide diversity, neutrality, and population structure. Coalescent simulations were performed to model domestication bottlenecks.	[66]

## Data Availability

Not applicable.

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
