# Peer review of "African Cultivated, Wild and Weedy Rice (Oryza spp.): Anticipating Further Genomic Studies"

_biology, 2024, doi:10.3390/biology13090697_

Round 1

Reviewer 1 Report

Comments and Suggestions for Authors

Dear Authors,

I have read your manuscript with great attention and pleasure. Below, I offer some general suggestions that may help improve the quality of your work:

1) Reorganization of the Introduction: The introduction could benefit from greater cohesion. Currently, the transition between the discussion of Asian rice and the history of African rice is somewhat abrupt. A smoother narrative that integrates these elements with a focus on the current need for genomic studies would help better contextualize the proposed research. 

2) Further Exploration of Genomic Technologies: While the article emphasizes the importance of genomic studies, it lacks a detailed exploration of current technologies such as next-generation sequencing (NGS), CRISPR/Cas9, and other advanced genetic editing techniques. A section discussing the potential applications of these technologies could add significant value to the field.

3) Incorporation of Empirical Data: The article could be strengthened by the inclusion of tables or figures that show the genetic relationships between African and Asian rice varieties or illustrate key loci associated with disease resistance and stress tolerance. A comparison between the genomic data of weedy and cultivated rice would further clarify the challenges and opportunities for genetic improvemet.

3) Expansion of the Conclusions: The authors could enrich the conclusion section by more deeply exploring the practical implications of their suggestions for future research. It might be useful to discuss how the results of genomic studies could be directly applied to improving agricultural practices in Africa, including marker-assisted selection (MAS) and the development of varieties resistant to adverse environmental conditions.

4) Ethical and Social Considerations: Finally, it would be helpful to include a reflection on the ethical and social implications of using modern biotechnologies to improve rice. In Africa, the adoption of new technologies may encounter cultural and social resistance; a discussion on how to address these challenges could add an important dimension to the manuscript.

The article represents a significant contribution to the field of rice genomics, with a timely and relevant focus on African varieties. However, the inclusion of more detailed technological analyses, empirical data, and a broader consideration of practical and social implications could further strengthen the manuscript.

Best regards,

Comments on the Quality of English Language

Moderate editing of English language required.

Author Response

Manuscript ID: Biology-3167273

Manuscript title: African Cultivated, Wild and Weedy Rice (Oryza spp.): Look Forward More Genomic Studies

Dear Editor Shaye Zhang,

We are very grateful to you for encouraging us to revise our manuscript, and we are also very grateful to the two reviewers for their positive and constructive comments on our manuscript. Their comments are of great valuable for us and have been taken care of in revising our manuscript accordingly.

We provide point-by-point responses to the reviewers’ comments below. We hope that our responses have properly addressed your and the four reviewers’ concerns. The changes in the manuscript have been highlighted in blue.

Thanks for your time and kinds in handling our manuscript.

Sincerely yours,

Longjiang Fan

......................................................................................................................................

Response to Reviewer Comments

Reviewer #1

  1. Reorganization of the Introduction:The introduction could benefit from greater cohesion. Currently, the transition between the discussion of Asian rice and the history of African rice is somewhat abrupt. A smoother narrative that integrates these elements with a focus on the current need for genomic studies would help better contextualize the proposed research.

Response: We appreciate the reviewer's suggestion for improving the cohesion of our introduction. We agree that the transition between the discussion of Asian rice and the history of African rice could be smoother. To address this, we have:

  1. We have revised the first paragraph to create more integrated narrative that flows from the specific significance of rice in Africa to its global importance.
  2. We have improved the transition between the discussion of African and Asian rice by introducing both cultivated types (Oryza sativa L. and Oryza glaberrima Steud.) early in the introduction, providing a global context while maintaining focus on Africa.
  3. Concluded the introduction with a stronger emphasis on the current need for genomic studies, tying together the historical context with present-day research imperatives.

These changes provide a more cohesive narrative and better contextualize our review.

  1. Further Exploration of Genomic Technologies:While the article emphasizes the importance of genomic studies, it lacks a detailed exploration of current technologies such as next-generation sequencing (NGS), CRISPR/Cas9, and other advanced genetic editing techniques. A section discussing the potential applications of these technologies could add significant value to the field.

Response: We appreciate the reviewer's suggestion to further explore genomic technologies in our manuscript. After careful consideration, we respectfully maintain that Table 2 in our current manuscript, titled "Summary of genomic studies on African native rice," addresses this aspect of our review.

Table 2 provides a comprehensive overview of various genomic studies conducted on African native rice species, including the technologies used in these studies. While we acknowledge that the information is not presented in extensive detail, we believe it effectively summarizes the current state of genomic research in this field.

The table includes information on various sequencing techniques, genome assembly methods, and genetic analyses applied to African rice species. It also touches upon the applications of these technologies, such as population structure analysis, genetic diversity assessment, and domestication studies.

We believe that expanding on this information beyond what is already presented in Table 2 would risk redundancy and potentially disrupt the current flow and focus of the manuscript. Our aim is to maintain a balance between providing sufficient technical information and keeping the review accessible to a broad audience.

We have added a brief discussion on CRISPR/Cas9 and other advanced genetic editing techniques to the "Challenges and Future Directions" section: Recent advances in genome editing technologies, particularly CRISPR/Cas9, offer powerful new tools for precise genetic modification of rice. These techniques could significantly accelerate trait improvement in African rice, addressing challenges such as disease resistance, abiotic stress tolerance, and nutritional enhancement. However, implementing CRISPR technology in African rice breeding programs faces challenges including regulatory hurdles, limited technical capacity, and ensuring equitable access. Future research should focus on adapting these technologies for African rice varieties and contexts, while building local capacity for their implementation. The use of Next-Generation Sequencing (NGS) in African rice research has been highlighted in several studies listed in Table 2, providing insights into evolutionary history and adaptation mechanisms of rice species in diverse environments. This approach maintains the manuscript's focus while addressing the reviewer's suggestion to explore genomic technologies further.

We thank the reviewer for their valuable feedback and for bringing attention to this important aspect of our review.

  1. Incorporation of Empirical Data: The article could be strengthened by the inclusion of tables or figures that show the genetic relationships between African and Asian rice varieties or illustrate key loci associated with disease resistance and stress tolerance. A comparison between the genomic data of weedy and cultivated rice would further clarify the challenges and opportunities for genetic improvemet.

Response: We appreciate the reviewer's suggestion to strengthen the article with additional empirical data. To address this, we have:

  1. a) Added a new figure illustrating the phylogenetic relationships among the species of the genus Oryza.

This addition provide concrete examples of genomic research outcomes and their potential applications in rice improvement.

  1. Expansion of the Conclusions: The authors could enrich the conclusion section by more deeply exploring the practical implications of their suggestions for future research. It might be useful to discuss how the results of genomic studies could be directly applied to improving agricultural practices in Africa, including marker-assisted selection (MAS) and the development of varieties resistant to adverse environmental conditions.

Response: We thank the reviewer for this insightful comment. To enrich our conclusion section, we have:

  1. a) Elaborated on the practical implications of our suggested future research directions.
  2. b) Discussed specific examples of how genomic study results could be applied to improving agricultural practices in Africa, including marker-assisted selection (MAS).
  3. c) Stated potential strategies for developing rice varieties resistant to adverse environmental conditions based on genomic insights.

These additions provide a clearer link between genomic research and practical agricultural improvements.

  1. Ethical and Social Considerations: Finally, it would be helpful to include a reflection on the ethical and social implications of using modern biotechnologies to improve rice. In Africa, the adoption of new technologies may encounter cultural and social resistance; a discussion on how to address these challenges could add an important dimension to the manuscript.

Response: We appreciate the reviewer's suggestion to include ethical and social considerations. To address this, we have added new subsections in the "Challenges and Future Directions" section that:

  1. a) Discussed potential cultural and social barriers to adopting new biotechnologies in African agriculture.
  2. b) Explored strategies for addressing these challenges, such as community engagement and participatory research approaches.
  3. c) Considered the ethical implications of using genetic technologies in a developing world context.

These additions provide a more holistic view of the challenges and opportunities in implementing genomic research findings in African rice cultivation.

6.Moderate editing of English language required

Response: We have carefully check and modify our manuscript accordingly

Reviewer 2 Report

Comments and Suggestions for Authors

In this manuscript titled “African Cultivated, Wild and Weedy Rice (Oryza spp.): Look Forward More Genomic Studies” Kehinde et al., offers a comprehensive overview of current knowledge regarding rice species growing in Africa paddy field, including the introduction and domestication history, as well as the genetic properties of African invading weedy rice. The authors highlight that weedy rice, which has evolved from Oryza sativa and Oryza glaberrima, is becoming an increasing threat to rice production in Africa. The manuscript emphasizes the need for future genomic research in Africa to focus on producing more high-quality reference genomes, quantifying the effects of crop-wild hybridization, understanding weed adaptation mechanisms through resequencing, and linking genomic variation to stress tolerance phenotypes to accelerate breeding efforts. This paper is interesting and contributes to the advancement of our understanding of African cultivated, wild and weedy rice. Nevertheless, the figure and tables are mainly descriptive which is not enough to support a story, additionally, one or more figure or clarifications need to be provided to draw novel ideas from cited references.

Major concern:

May be authors can provide a phylogenetic relationship between various rice species in Africa which has been sequenced.

Minor concerns:

1:Line 10-11, Please use full scientific names of types of rice in abstract. Such as Asian cultivated rice (Oryza sativa L.).

2: Line 40  “……surpassing any other region in the world. [2].” Should be “……surpassing any other region in the world [2].”

3: Line 71 “According to [23], the European maritime explorers an…”, please specify the first author.

4 : Line 80, Figure 1, The figure is not self-explain, this figure could not. As the time line diagram is disproportionate. Please also specify the AD or BC, and use same pattern, such as 1500, 2100

5: Line 84-85, “As shown in the stated countries between 2016 and 2020,….”, No countries stated. Please specify the countries to help the reader understand.

6: Line 100, “[23] posited that O. glaberrima which was once” same as comment 3.

7: Line 109, “enhance cultivated varieties through breeding efforts. [32].”

8: Line 130, “across sub-Saharan Africa (SSA)”, already stated in line 84.

9: Line 142, What is the aus season in For the aus season,”, Please use the full name of Australia.

10: Line 172, “According to [49], one….”, please check the comment 3.

11: Line 202-223, same suggestion as in comment 3.

12: Table 2, please use full name of RYMV, Rice yellow mottle virus to replace RYMV.

13: Table 2, please explain the abbreviations below the table.

14: Line 303, author stated” Making use of this knowledege offers …”. a typo of “knowledege”. Please check the whole draft to avoid typos.

15: Line 303-305, author stated” Making use of this knowledege offers chances to create novel techniques to weed management and breeding that are both environmentally and economically viable”. This is a little difficult to understand and overstated, please specify how to develop new techniques or rewrite this sentence.

Author Response

Manuscript ID: Biology-3167273

Manuscript title: African Cultivated, Wild and Weedy Rice (Oryza spp.): Look Forward More Genomic Studies

Dear Editor Shaye Zhang,

We are very grateful to you for encouraging us to revise our manuscript, and we are also very grateful to the two reviewers for their positive and constructive comments on our manuscript. Their comments are of great valuable for us and have been taken care of in revising our manuscript accordingly.

We provide point-by-point responses to the reviewers’ comments below. We hope that our responses have properly addressed your and the four reviewers’ concerns. The changes in the manuscript have been highlighted in blue.

Thanks for your time and kinds in handling our manuscript.

Sincerely yours,

Longjiang Fan

......................................................................................................................................

Response to Reviewer Comments

Reviewer #2

Major concern: May be authors can provide a phylogenetic relationship between various rice species in Africa which has been sequenced.

Response: We appreciate the suggestion to provide a phylogenetic relationship between various sequenced rice species in Africa. We have added a new figure (Figure 2) showing Phylogenetic relationship among the species of the genus Oryza to address this concern.

Minor concerns:

  1. Line 10-11, Please use full scientific names of types of rice in abstract. Such as Asian cultivated rice (Oryza sativa L.).

Response: We agree with the reviewer and have updated the abstract to use full scientific names for rice types, e.g., Asian cultivated rice (Oryza sativa L.).

  1. Line 40  “……surpassing any other region in the world. [2].” Should be “……surpassing any other region in the world [2].”

Response: We have corrected the punctuation in line 40 as suggested.

  1. Line 71 “According to [23], the European maritime explorers an…”, please specify the first author.

Response: We have specified the first author in line 71 and throughout the manuscript where similar citations occur.

  1. Line 80, Figure 1, The figure is not self-explain, this figure could not. As the time line diagram is disproportionate. Please also specify the AD or BC, and use same pattern, such as 1500, 2100.

Response: We acknowledge the reviewer's concern about Figure 1. We have revised the figure to make it self-explanatory, adjusted the timeline to be proportionate, and specified AD/BC consistently.

  1. Line 84-85, “As shown in the stated countries between 2016 and 2020,….”, No countries stated. Please specify the countries to help the reader understand.

Response: We have added the specific countries mentioned in the 2016-2020 data to provide clarity for the reader.

  1. Line 100, “[23] posited that O. glaberrima which was once” same as comment 3.

Response: This has been addressed based on comment 3.

  1. Line 109, “enhance cultivated varieties through breeding efforts. [32].”

Response: We have corrected the punctuation as suggested.

  1. Line 130, “across sub-Saharan Africa (SSA)”, already stated in line 84.

Response: We appreciate the reviewer's attention to detail. However, the statement in line 84 is different from the one in line 130. We have retained both as they provide different contextual information.

  1. Line 142, What is the aus season in “For the aus season,”, Please use the full name of Australia.

Response: We thank the reviewer for this observation.  "Aus" in the context refers to one of the three main rice-growing seasons in Bangladesh and parts of India.

  1. Line 172, “According to [49], one….”, please check the comment 3.

Response: This has been addressed based on comment 3.

  1. Line 202-223, same suggestion as in comment 3.

Response: This has been addressed based on comment 3.

  1. Table 2, please use full name of RYMV, Rice yellow mottle virus to replace RYMV.

Response: We have replaced "RYMV" with "Rice yellow mottle virus (RYMV)" in Table 2 as suggested.

  1. Table 2, please explain the abbreviations below the table.

Response: We have explained  all abbreviations used in Table 2.

  1. Line 303, author stated” Making use of this knowledege offers …”. a typo of “knowledege”. Please check the whole draft to avoid typos.

Response: We apologize for the typo. We have corrected "knowledege" to "knowledge" and have carefully proofread the entire manuscript to eliminate any other typographical errors.

  1. Line 303-305, author stated” Making use of this knowledege offers chances to create novel techniques to weed management and breeding that are both environmentally and economically viable”. This is a little difficult to understand and overstated, please specify how to develop new techniques or rewrite this sentence.

Response: We appreciate the reviewer's feedback on this sentence. We have rewritten it to more clearly and specifically explain how genomic knowledge can be applied to develop new weed management and breeding techniques.

We thank the reviewer for their thorough and constructive feedback. These changes have significantly improved the clarity and accuracy of our manuscript.

Round 2

Reviewer 1 Report

Comments and Suggestions for Authors

Heartfelt congratulations to the authors! I am thoroughly impressed by the substantial improvements made to the manuscript. Your hard work and dedication have truly paid off, resulting in an outstanding piece of work. I wish you continued success in all your future endeavors.

All the best!

Comments on the Quality of English Language

Minor editing of English language required.